spectroscopy/computational chemistry/analytical chemistry

losartan, amlodipine besylate, second-derivative synchronous spectrofluorimetry, high-performance liquid chromatography and fluorimetric detection

**Author for correspondence:**
Shereen Shalan
e-mail: shereenshalan@yahoo.com

This article has been edited by the Royal Society of Chemistry, including the commissioning, peer review process and editorial aspects up to the point of acceptance.

# Simultaneous evaluation of losartan and amlodipine besylate using second-derivative synchronous spectrofluorimetric technique and liquid chromatography with time-programmed fluorimetric detection

## Shereen Shalan and Jenny Jeehan Nasr

Department of Pharmaceutical Analytical Chemistry, Faculty of Pharmacy, Mansoura University, Mansoura 35516, Egypt

SS, 0000-0002-1468-1367; JJN, 0000-0002-2423-7315

This study is concerned with two sensitive, fast and reproducible approaches; namely, second-derivative synchronous fluorimetry (method I) and reversed phase high-performance liquid chromatography with fluorimetric detection (method II) for synchronized evaluation of losartan (LOS) and amlodipine besylate (AML). Method I is based on measuring second-derivative synchronous fluorescence spectra of LOS and AML at $\Delta\lambda = 80$ nm in water. The experimental factors influencing the synchronous fluorescence of the considered compounds were sensibly adjusted. The chromatographic analysis was executed on a Nucleodur MN-C18 column of dimensions; $250 \times 4.6$ mm i.d. and 5 µm particle size). The fluorimetric detection was time-programmed at $\lambda em = 440$ nm for AML (0.0–7.5 min) and at $\lambda em = 400$ nm for LOS (7.5–10 min) after excitation at $\lambda ex = 245$ nm. The mobile phase is a blend of acetonitrile with 0.02 M phosphate buffer in a proportion of 45 : 55, pH 4.0, pumped using a flow rate of 1 ml min$^{-1}$. The calibration plots were established to be 0.1–4.0 µg ml$^{-1}$ for both drugs in method I and 0.05–4.0 µg ml$^{-1}$ for both drugs in method II. The study was extended to the evaluation of the two drugs in their co-formulated tablets.

# 1. Introduction

Losartan potassium (LOS) (2-*n*-butyl-4-chloro-5-hydroxymethyl-1-[2′-(1H-tetrazol-5-yl) (biphenyl-4-yl) methyl] imidazole) is a highly effective antihypertensive agent acting by antagonizing angiotensin II receptor (figure 1) [1]. The active form is taken and is partly transformed to an active metabolite that causes elongated pharmacological impact of the drug. Losartan therapeutic effect and its renal and hypotensive effects appear analogous to those of angiotensin-converting enzyme inhibitors [1].

Amlodipine (AML) (3-ethyl-5-methyl (4RS)-2-[(2-aminoethoxy) methyl]-4-(2- chlorophenyl)-methyl-1-ihydropyridine-3, 5-dicarboxylate benzenesulfonate) is a long-term calcium channel blocker employed as a hypotensive drug [2] (figure 1).

Review of the literature disclosed that spectrophotometric and chromatographic techniques have been used for evaluation of LOS [3–6], and AML [7–10] in solitary and multicomponent dosage forms or in biological fluids. However, there are few reported techniques used for concurrent determination of LOS and AML, such as spectrophotometry [11–15], high-performance liquid chromatography (HPLC) [16–19] and HPTLC [20].

The target of the current work is to set up and progress two new, sensitive and selective methods for synchronized evaluation of LOS and AML either in pure form or in pharmaceutical formulations employing derivative synchronous fluorescence spectroscopy (DSFS) (method I) and LC with fluorimetric detection (method II). In authors' information, neither synchronous spectrofluorimetry nor HPLC with fluorimetric detection were previously used for the synchronized analysis of LOS and AML in their co-formulated dosage forms.

Synchronous fluorescence spectroscopy (SFS) has a few focal points over routine fluorescence spectroscopy, comprising simple spectra, low interference in addition to high selectivity [21]. As a result of its sharp and fine spectrum, SFS functions as a simple and real technique for receiving records for measurable estimation in a solitary measurement [22]. The blend of SFS with derivatives is more favourable than routine emission band regarding sensitivity, since the size of the signal in derivative is inversely related to the thickness of band of original spectrum [23,24].

Lately, derivative synchronous fluorimetry is employed for estimation of many combinations in formulations and biological fluids. For example, binary mixtures of topiramate and levetiracetam [25], itopride and domperidone [26], nebivolol and amlodipine [27], desloratadine and montelukast sodium [28], AML and valsartan [29], adapalene acidic and oxidative products [30], methocabamol and ibuprofen [31] have been estimated using this technique.

# 2. Experimental methods

## 2.1. Materials

LOS and AML pure analytical standards were obtained from Sigma-Aldrich (St Louis, MO, USA) and used as obtained. Losar-A® tablets (Unichem Pharmaceuticals, Himachal Pradesh, India) considered to have 50 mg LOS and 5 mg AML in proportion 10 : 1 (Batch # 08455) were purchased from India.

## 2.2. Reagents

All solvents and reagents were of analytical reagent grade. Methanol and acetonitrile (HPLC grade) were purchased from Merck, Darmstadt, Germany. Acetate buffer was prepared by blending suitable amounts of acetic acid (0.2 M) with sodium acetate (0.2 M) in pH range of 3.6–5.6. Borate buffer (pH 5.5–13) was prepared by blending suitable amounts of boric acid (0.02 M) with sodium hydroxide (0.2 M). Dipotassium hydrogen phosphate was purchased from Sigma-Aldrich, Seelze, Germany. Phosphoric acid was from Riedel-deHaën (Seelze, Germany).

## 2.3. Apparatus

Spectrofluorimetric data were measured using a luminescence spectrometer of Perkin-Elmer model LS 45 (UK), containing a xenon arc lamp of 150 W, grating monochromators for excitation and emission, and a recorder of Perkin-Elmer. The used slit width for both monochromators was 10 nm. A quartz cuvette of 1 cm was used. Fluorescence Data Manager (FLDM) software was used for recording second-derivative spectra using a number of points of 99. The fluorescence intensities of the second-derivative spectra were measured at 260 and 380 nm for LOS and AML, respectively.

losartan potassium      amlodipine besylate

veratic acid

**Figure 1.** Structural formulae of losartan potassium, amlodipine besylate and veratic acid (internal standard).

A Shimadzu Prominence HPLC system (Shimadzu Corporation, Japan) with an LC-20 AD pump was used to carry out chromatographic analyses. DGU-20 A5 degasser, CBM-20A interface and fluorimetric detector (FXL-10A) with 20 µl injection loop were used. The column used was a Nucleodur MN-C18 column of dimensions 250 × 4.6 mm i.d. and particle size of 5 µm) purchased from Macherey-Nagel, Düren, Germany.

pH adjustments were made using pH meter (Model pHS-3C) obtained from Shanghai Leici Instruments Factory, China. A sonicator BHA-180 T obtained from Abbotta Corporation in the USA was used.

## 2.4. Development of HPLC method

The chromatographic analysis was carried out on a C18 column. The column was used at room temperature. The HPLC system was washed daily with 50 ml of 1 : 1 methanol : water mixture to eliminate the mobile phase; this led to reproducibility of the analytical column. A mobile phase consisted of a blend of acetonitrile with 0.02 M phosphate buffer at a proportion of 45 : 55 with pH 4.0 controlled using sodium hydroxide (0.2 M) or orthophosphoric acid (0.2 M), with sonication of the mobile phase for 30 min. Then filtration of the mobile phase was done through a membrane filter of pore size 0.45 µm purchased from Sartorius-Stedim, Goettingen, Germany. The separation was done using time-programmed fluorimetric detection at $\lambda$em = 440 nm for AML (0.0–7.5 min) and at $\lambda$em = 400 nm for LOS (7.5–10 min) after excitation at $\lambda$ex = 245 nm.

## 2.5. Standard solutions

Stock solutions of each of LOS and AML prepared by dissolving 10.0 mg of each compound in 100 ml volumetric flask and completed to the volume with methanol were additionally diluted with water (method I), and were additionally diluted by mobile phase (in method II) as suitable. The stability of standard solutions was found to be stable for 10 days if refrigerated.

# 3. Suggested procedures

## 3.1. Construction of calibration graphs of second-derivative synchronous fluorescence spectroscopy method (method I)

Aliquots of LOS and AML standard solutions throughout the working range of concentration as mentioned in table 2 (0.1–4.0 µg ml$^{-1}$ for both LOS and AML) were conveyed into a sequence of 10 ml volumetric flasks, 2 ml of 1% sodium dodecyl sulfate (SDS) solution was added, then diluted

with water and blended well. The SFS of the samples were performed at a constant difference of wavelength $\Delta\lambda = 80$ nm. The second-derivative fluorescence spectra (SDFS) of LOS and AML were calculated from standard synchronous spectra. The second-derivative spectra peak heights were determined at 260 and 380 nm for LOS and AML, correspondingly. A blank experimentation was executed in the same time. The second-derivative method peaks amplitude was graphed against the ultimate concentration of the studied compounds ($\mu g\ ml^{-1}$) to obtain the calibration curve. On the other hand, the equivalent regression equations were inferred.

## 3.2. Construction of calibration graphs of HPLC method (method II)

Working solutions containing $0.05-4.0\ \mu g\ ml^{-1}$ for each LOS and AML were set by successive dilutions of the stock solutions. To each flask, $2\ \mu g\ ml^{-1}$ veratic acid was added as internal standard (IS), then mobile phase was added to dilute to final volume. Samples of $20\ \mu l$ volume were injected in triplicate and chromatographed under prescribed chromatographic conditions. Peak area ratios (drug/IS) of each studied drug were plotted versus the concentrations of drug in $\mu g\ ml^{-1}$. On the other hand, the equivalent regression equations were inferred.

## 3.3. Method for the laboratory-prepared mixtures

Aliquots of LOS and AML standard solutions in the therapeutic proportion of 10 : 1 were moved into a series of 10 ml volumetric flasks. Then, the steps described under Construction of calibration curves for the second-derivative synchronous fluorescence spectroscopy (SDSFS) method (method I) or for the HPLC method (method II) were performed. On the other hand, the equivalent regression equations were inferred.

# 4. Applications

Ten tablets were weighed and finely powdered. A weighed amount of the powder corresponding to 50 mg LOS and 5 mg of AML (in the therapeutic proportion of 10 : 1) was moved to a 100 ml volumetric flask then extracted with methanol. After sonication for 30 min, filtration was done. Aliquots covering the working concentration range were transmitted to 10 ml volumetric flasks. The recommended procedures in §§3.1 and 3.2 were performed. The nominal content of the tablets were determined either from a previously plotted calibration graph or using the corresponding regression equation.

# 5. Results and discussion

LOS and AML show innate fluorescence with determined wavelengths of 355 and 440 nm, using excitation at 267 and 239 nm for LOS and AML, correspondingly. Overlapping of both spectra of excitation and emission of LOS and AML occurred (figure 2). Thus, the usage of routine fluorescence method for the simultaneous estimation of LOS and AML is difficult, especially if it is required to estimate these compounds in their co-formulated dosage form.

The standard synchronous spectra for LOS and AML were first recorded to develop the SDS spectra. Figure 3 demonstrates the SF spectra of various concentrations of AML at 365 nm in presence of unchanged concentration of LOS ($2.0\ \mu g\ ml^{-1}$). It is obvious that there is still an overlap in the SF spectra of both drugs. Hence, the second-derivative synchronous fluorescence spectroscopy (SDSFS) method was selected for concurrent estimation of LOS and AML together in their dosage forms. Spectra of LOS and AML were greatly differentiated using SDSFS with zero-crossing method (figures 4 and 5). Under the states of the test, the two peaks showed up at 260 and 380 nm for LOS and AML, respectively.

## 5.1. Optimization of reaction condition

Distinctive test factors influencing the execution of the presented technique were deliberately studied and enhanced. These parameters were altered separately, with others were held consistent. These parameters comprised choice of $\Delta\lambda$, pH, kind of the diluting solvent, and stability time.

### 5.1.1. Choice of ideal $\Delta\lambda$

The ideal $\Delta\lambda$ value is vital for executing the SF scanning procedure in the view of its separation, sensitivity and characteristics. It may specifically affect synchronous spectral outline, width of band

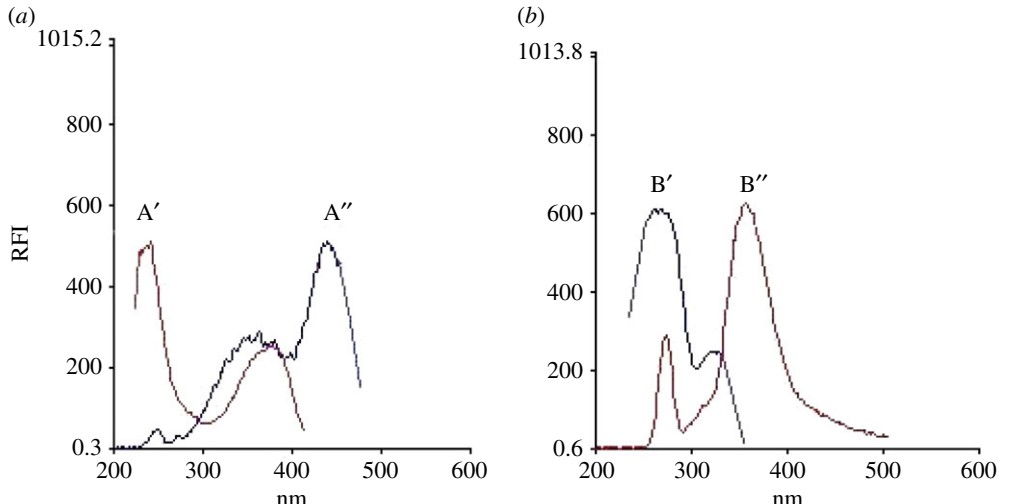

**Figure 2.** Native fluorescence of (*a*) AML and (*b*) LOS. A′ and A″ are the emission and excitation spectra of AML (2 μg ml$^{-1}$). B′ and B″ are the emission and excitation spectra of LOS (2 μg ml$^{-1}$). (RFI, relative fluorescence intensity)

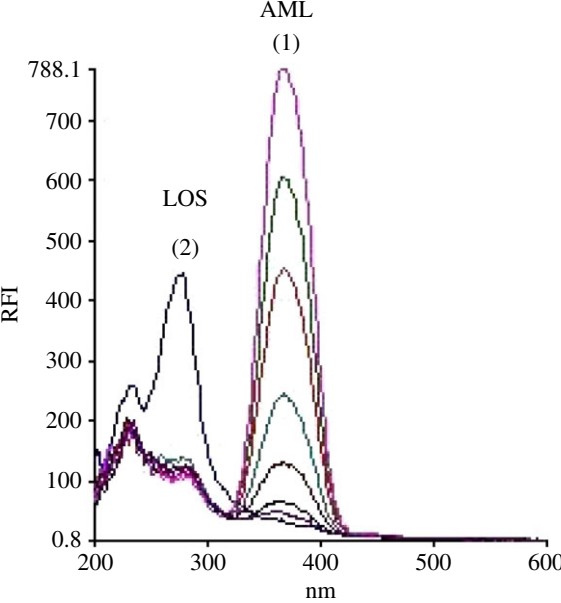

**Figure 3.** SF spectra of different concentrations of AML at 365 nm in the presence of constant concentration of LOS (2.0 μg ml$^{-1}$).

and signal magnitude. For this object, an extensive choice of Δλ ranging from 20 to 120 nm was inspected. It was found that Δλ lower than 80 nm caused the bands outlines to be uneven and noisy with weak fluorescence amplitudes. Alternatively, Δλ greater than 80 nm caused overlap of both peaks causing bad resolution. Consequently, Δλ of value 80 was selected as ideal for resolution of LOS and AML combinations, because it caused two separate peaks with right even outlines and lessened the bands interference resulting from every compound in the mixture. Figures 6 and 7 demonstrate these outcomes.

In figures 6 and 7, the three-dimensional synchronous spectra of LOS and AML are depicted by means of surface projection and contour plots, wherever synchronous bands at gradual increases of Δλ were recorded then graphed.

### 5.1.2. Impact of various organized media

The impact of addition of different organized media on the synchronous fluorescence intensities of the two drugs was considered, using SDS as an anionic surfactant, CTAB as a cationic surfactant,

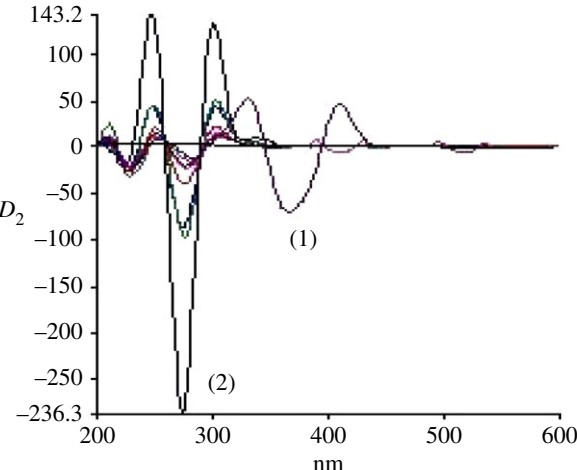

**Figure 4.** Second-derivative synchronous spectra of different concentration of LOS (0.1–4.0 μg ml⁻¹) in the presence of AML (2 μg ml⁻¹).

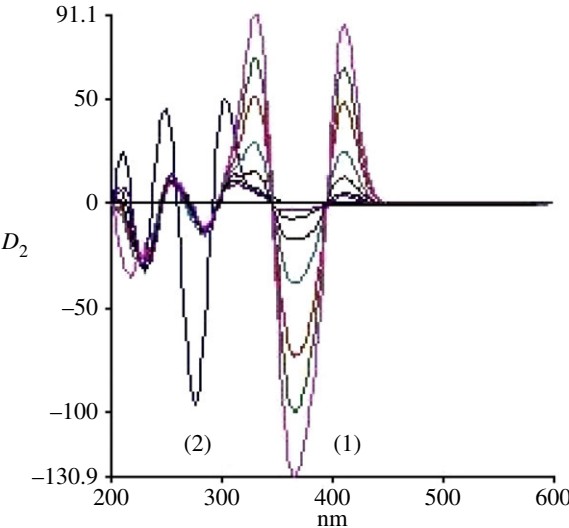

**Figure 5.** Second-derivative synchronous spectra of different concentration of AML (0.1–4.0 μg ml⁻¹) in the presence of LOS (2 μg ml⁻¹).

Tween-80 as a non-ionic surfactant, methyl cellulose and HP-β-CD. To the aqueous solution of the drugs mixture, 2 ml of every surfactant (1.0% w/v) was added (final concentration 2.0 μg ml⁻¹ for LOS and 1.0 μg ml⁻¹ for AML). It was found that only SDS produced a substantial rise in the relative fluorescence intensity; therefore, SDS was chosen as the fluorescence promoter for studied drugs as shown in electronic supplementary material, figure S1.

### 5.1.3. Impact of the SDS volume

The impact of the SDS volume on the FI was studied through increasing volumes of 1% SDS. The results obtained revealed that by increasing the volumes of SDS solutions, there is increasing in the fluorescence intensities up till 2.0 ml after which additional rise in volume resulted in no additional rise in relative fluorescence intensity (RFI). Therefore, 2.0 ml of SDS solution of concentration 1% w/v was selected to be the optimal volume for each of LOS and AML.

### 5.1.4. Impact of the SDS concentration

The effect of different concentration of SDS was studied (0.25–1.5% w/v). The results revealed that increasing SDS concentration led to increasing fluorescence intensities up to 1.0% SDS, after which

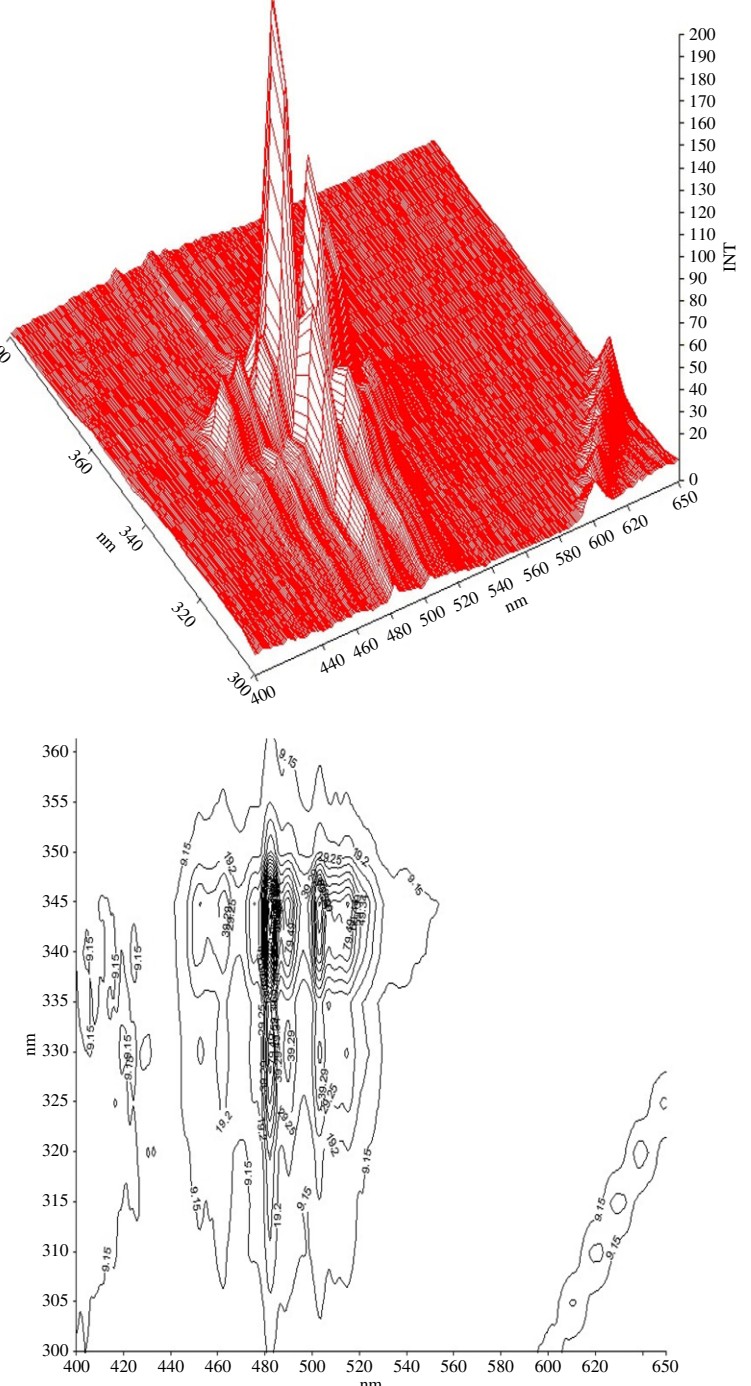

**Figure 6.** Effect of choice of $\Delta\lambda$ on AML (2 $\mu$g ml$^{-1}$). (INT, intensity)

additional rise in concentration resulted in no additional rise in RFI. So, 1.0% w/v SDS solution was chosen as the optimal concentration for both studied drugs.

### 5.1.5. Impact of diluting solvent

The impact of various diluting solvents on the RFI of LOS and AML in presence of SDS was considered using water, acetonitrile methanol, dimethyl sulfoxide and dimethyl formamide. The results obtained showed that water was the optimal diluting solvent due to giving the maximum RFI and the minimum blank reading. In the case of using methanol and acetonitrile, an obvious and sharp drop in the RFI occurred. This impact is ascribed to their denaturizing impact on micelles, as alcohols of short-chain like methanol are solubilized chiefly in the water phase and influence the micelle-formation procedure

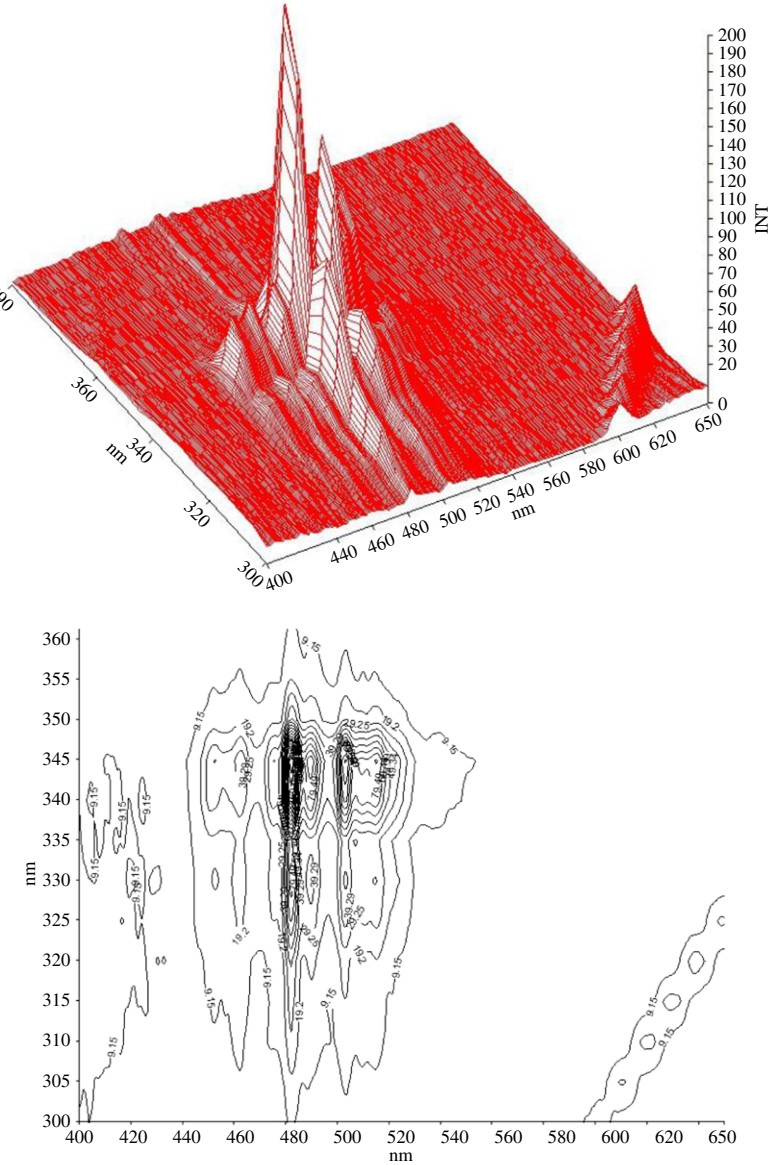

**Figure 7.** Effect of choice of $\Delta\lambda$ on LOS (2 $\mu$g ml$^{-1}$).

via altering the characteristics of the solvent. Furthermore, adding those organic solvents decreases the dimensions of micelles, but with an advanced collapse of the surfactant aggregate at very high concentrations [32]. Both dimethyl formamide and dimethyl sulfoxide reduced the fluorescence intensities of LOS and AML, as they commenced an intersystem crossing process (similar to the heavy atom effect) [33] as illustrated in electronic supplementary material, figure S2.

### 5.1.6. Choice of optimal pH

The pH impact on the SF intensities of the binary mixture was considered using various buffers to cover all the pH range, where acetate buffer and borate buffer were used in the pH range of 3.5–13. No effect was observed on the synchronous fluorescence intensity of LOS and AML upon using different pH values. Therefore, no buffer was used in the study, as shown in electronic supplementary material, figure S3.

## 5.2. HPLC method

The developed HPLC method allows the estimation of LOS and AML with good resolution in a reasonable time as shown in figure 8 (LOS ($t$R = 9.07 min) and AML ($t$R = 6.01 min)). Furthermore, the suggested method shows good sensitivity, precision and accuracy with limit of detection (LOD)

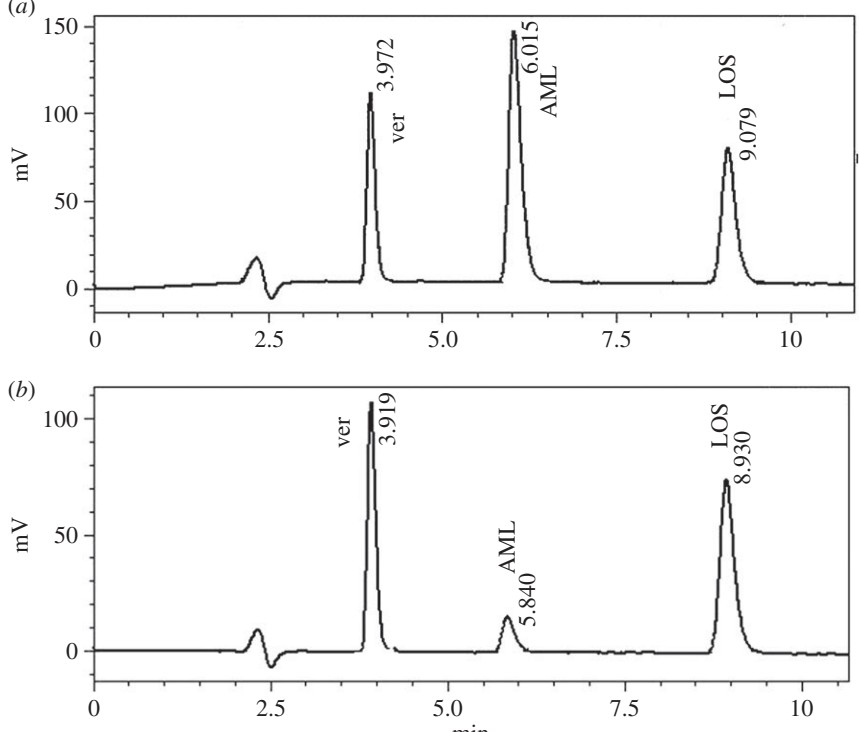

**Figure 8.** (*a*) Typical chromatogram of 2 μg ml$^{-1}$ veratic acid (IS), 2 μg ml$^{-1}$ AML and 4 μg ml$^{-1}$ LOS under described chromatographic condition. (*b*) Chromatogram of Losar-A$^{®}$ tablet 0.4 μg ml$^{-1}$ AML and 4 μg ml$^{-1}$ LOS in the presence of 2 μg ml$^{-1}$ (IS) under described chromatographic condition.

0.018 μg ml$^{-1}$ LOS and 0.048 μg ml$^{-1}$ AML. The suggested method can be applied for examination of the quality of the compounds under study, either *per se* or in their co-formulations.

### 5.2.1. Choice and optimization of chromatographic conditions

The optimization of the mobile phase constituents was carried out to obtain a highly selective and sensitive method through a brief period of time in order to attain distinct regular peaks. Table 1 shows the outcomes of the optimization of various chromatographic settings influencing the decisiveness and separation of LOS and AML. The constituents of the mobile phase were modified so as to study the probability of varying the chromatographic selectivity. These modifications involved variations of kind and proportion of organic modifier, pH and flow rate.

### 5.2.2. Selection of suitable wavelength and time programme adjustment

The fluorescence detector signals of LOS and AML were carefully considered and the top emission wavelengths were established at 400 and 440 nm after excitation at 245 nm for LOS and AML, correspondingly giving the best sensitivity. The programmed fluorescence detection was adjusted for allowing sensitive determination of both LOS and AML concurrently. For AML, it was recorded at 440 nm within 0–7.5 min, while LOS was recorded at 400 nm within 7.5–10 min.

Different organic modifiers like acetonitrile, *n*-propanol and methanol were studied to choose the most suitable one for the study. Acetonitrile proved to be the best, giving the maximum number of theoretical plates and well-separated peaks within sensible retention times. The impact of varying the proportion of organic modifier was considered on the retention times and selectivity of the studied compounds.

Mobile phases having ratios of acetonitrile: 0.02 M phosphate buffer (50 : 50), (45 : 55) and (40 : 60) were considered. The mobile phase less than 40% acetonitrile resulted in retardation of the peaks of both drugs, while mobile phase having ratios higher than 50% of acetonitrile resulted in low resolution. Hence, a ratio of mobile phase comprising acetonitrile : 0.02 M phosphate buffer at a proportion of 45 : 55 proved to be the greatest due to providing the optimal number of theoretical plates and sharp symmetrical peak.

The pH impact of mobile phase on the retention times and the selectivity of the studied compounds were considered using mobile phases with different pH values which ranged from 3.0 to 5.0. The pH 4.0

**Table 1.** Optimization of experimental parameters affecting the performance of the proposed chromatographic method.

| parameter | no. of theoretical plates | | tailing factor | |
|---|---|---|---|---|
| | LOS | AML | LOS | AML |
| organic modifier ratio (45%) | | | | |
| acetonitrile | 8489 | 6207 | 1.302 | 1.421 |
| *n*-propanol | 5273 | 4492 | 1.441 | 1.452 |
| methanol | 6265 | 5297 | 1.517 | 1.511 |
| acetonitrile concentration (%) | | | | |
| 40 | 7612 | 5346 | 1.420 | 1.490 |
| 45 | 8523 | 6327 | 1.310 | 1.421 |
| 50 | 6689 | 5997 | 1.530 | 1.620 |
| pH | | | | |
| 3.5 | 7524 | 4691 | 1.650 | 1.560 |
| 4.0 | 8576 | 6314 | 1.310 | 1.413 |
| 4.5 | 6287 | 5672 | 1.642 | 1.570 |
| 5.0 | 5379 | 3298 | 1.571 | 1.710 |

**Table 2.** Performance data of the proposed methods.

| parameter | SDSF | | HPLC | |
|---|---|---|---|---|
| | LOS | AML | LOS | AML |
| concentration range ($\mu$g ml$^{-1}$) | 0.1 – 4.0 | 0.1 – 4.0 | 0.05 – 4.0 | 0.05 – 4.0 |
| LOD ($\mu$g ml$^{-1}$) | 0.004 | 0.009 | 0.006 | 0.015 |
| LOQ ($\mu$g ml$^{-1}$) | 0.014 | 0.028 | 0.018 | 0.048 |
| correlation coefficient (*r*) | 0.9999 | 0.9999 | 0.9999 | 0.9999 |
| slope | 67.42 | 34.12 | 0.326 | 0.0356 |
| intercept | 8.281 | 2.599 | −0.0063 | 0.519 |
| *Sy/x* | 0.175 | 0.1733 | 0.00122 | 0.005 |
| *Sa* | 0.099 | 0.097 | 0.0026 | 0.0025 |
| *Sb* | 0.052 | 0.046 | 0.0014 | 0.0014 |
| % RSD | 0.302 | 0.745 | 0.821 | 1.289 |
| % Er | 0.123 | 0.280 | 0.310 | 0.484 |

was the optimum, giving reasonable separation, sharp peak and the maximum number of theoretical plates. The pH 3.0 resulted in broadening of peak; however, pH 5.0 produced low sensitivity for the studied drugs.

For the purpose of choosing an appropriate internal standard, two different drugs were studied: veratic acid and ethopabate. Veratic acid proved to be the best one, producing a well-separated peak from each of the studied drugs.

# 6. Validation of method

The proposed methods validity was studied in expressions of linearity, ranges, accuracy, precision, limits of detection and limits of quantification.

**Table 3.** Statistical analysis of the results obtained by the proposed and comparison methods for pure samples of LOS and AML.

| parameter | LOS | | | AML | | |
|---|---|---|---|---|---|---|
| | SDSF | HPLC | comparison method [17] | SDSF | HPLC | comparison method [17] |
| % recovery | 99.84 | 99.5 | 99.15 | 99.79 | 97.17 | 100.31 |
| | 99.45 | 98.68 | 99.78 | 98.6 | 99.84 | 99.54 |
| | 100.01 | 101.09 | 100.01 | 99.16 | 98.75 | 99.62 |
| | 100.38 | 99.49 | | 100.74 | 99.65 | |
| | 99.90 | 99.73 | | 100.20 | 99.86 | |
| | 100.00 | 100.34 | | 100.01 | 100.93 | |
| | | 99.94 | | 99.91 | 99.78 | |
| $\overline{X}$ | 99.93 | 99.81 | 99.64 | 99.76 | 99.36 | 99.82 |
| ± s.d. | ± 0.302 | ± 0.820 | ± 0.445 | ± 0.743 | ± 1.280 | ± 0.423 |
| Student's $t$-value[a] | 0.346 (1.895) | 1.20 (1.86) | | 0.131 (1.860) | 0.646 (1.86) | |
| variance ratio $F$-value[a] | 1.473 (5.79) | 1.84 (5.79) | | 1.756 (5.14) | 3.026 (5.79) | |

[a]Tabulated $t$- and $F$-values at $p = 0.05$ are given in parentheses.

**Table 4.** Accuracy and precision data for LOS and AML using the proposed methods. Each result is the average of three separate determinations.

| parameter | concentration ($\mu$g ml$^{-1}$) | intra-day[a] | | inter-day[b] | |
| --- | --- | --- | --- | --- | --- |
| | | recovery (mean $\pm$ s.d.) | %Er | recovery (mean $\pm$ s.d.) | %Er |
| **SDSF method** | | | | | |
| LOS | 0.1 | 99.34 $\pm$ 0.91 | 0.50 | 100.05 $\pm$ 0.65 | 0.56 |
| | 0.5 | 98.96 $\pm$ 0.82 | 0.47 | 100.48 $\pm$ 1.31 | 1.23 |
| | 1.0 | 100.03 $\pm$ 1.21 | 0.39 | 99.56 $\pm$ 1.22 | 1.98 |
| AML | 0.5 | 99.47 $\pm$ 0.63 | 0.61 | 99.62 $\pm$ 0.77 | 0.62 |
| | 1.0 | 98.99 $\pm$ 1.61 | 0.59 | 100.34 $\pm$ 0.83 | 0.81 |
| | 2.0 | 99.49 $\pm$ 1.25 | 0.61 | 98.78 $\pm$ 0.39 | 0.37 |
| **HPLC method** | | | | | |
| LOS | 0.1 | 99.69 $\pm$ 0.91 | 0.79 | 99.94 $\pm$ 0.79 | 0.65 |
| | 0.5 | 100.42 $\pm$ 0.83 | 0.81 | 99.31 $\pm$ 0.34 | 0.32 |
| | 1.0 | 100.52 $\pm$ 0.72 | 0.71 | 100.62 $\pm$ 0.71 | 0.68 |
| AML | 0.5 | 99.88 $\pm$ 0.59 | 0.49 | 100.27 $\pm$ 1.42 | 1.37 |
| | 1.0 | 100.63 $\pm$ 0.87 | 0.53 | 99.69 $\pm$ 0.51 | 0.46 |
| | 2.0 | 98.99 $\pm$ 1.11 | 0.98 | 99.90 $\pm$ 0.68 | 0.63 |

[a]Intra-day: within the day.
[b]Inter-day: three consecutive days.

## 6.1. Linearity and range

The linearity graphs of the proposed methods were obtained. The calibration graphs presented a linear response between (second-derivative largeness values) in method I or ratios of peak area in method II and concentrations of the drug. The concentration ranges were established to be 0.1–4.0 $\mu$g ml$^{-1}$ for the two compounds, LOS and AML in method I; whereas in method II, the concentration ranges for LOS and AML were found to be 0.05–4.0 $\mu$g ml$^{-1}$.

The excellent estimate of correlation coefficients ($r > 0.9999$) in addition to lowered intercept values designate low scattering points around the calibration curves. The statistical results of the data [34] are presented in table 2.

## 6.2. Limit of quantification and limit of detection

The limit of quantification (LOQ) was determined by measuring the lowest concentration that can be measured according to ICH Q2(R1) recommendations [35], below which the calibration graph is nonlinear. Table 2 shows the results of LOD and LOQ of LOS and AML by the suggested methods.

## 6.3. Accuracy and precision

The results obtained by the suggested methods were matched with the results obtained from comparison method [17]. The proposed methods are more sensitive than the comparison method, which used the mobile phase 0.02% triethylamine in water and acetonitrile (60 : 40), at pH 2.5 on C18 column at 226 nm. The linearity ranges are 50–500 $\mu$g ml$^{-1}$ and 5–50 $\mu$g ml$^{-1}$ for LOS and AML, respectively. Statistical analysis of the results using Student's $t$-test and variance ratio $F$-test (table 3) revealed no significant difference between the performance of the proposed and comparison method according to accuracy and precision.

Accuracy and precision of inter-day and intra-day were evaluated for the proposed methods and the results are abridged in table 4. The small values of % relative standard deviation and % error prove that the proposed methods are accurate and precise.

**Table 5.** Determination of LOS and AML in co-formulated tablet (Losar-A®) using the proposed methods. Losar-A® tablets labelled to contain 50 mg of LOS and 5 mg of AML.

| parameter | ($\mu$g ml$^{-1}$) taken | | % recovery | | comparison method [17] | |
|---|---|---|---|---|---|---|
| | LOS | AML | LOS | AML | LOS | AML |
| SDSF method | 1.0 | 0.1 | 99.95 | 99.73 | 100.16 | 100.25 |
| | 2.0 | 0.2 | 99.82 | 100.81 | 100.09 | 99.66 |
| | 4.0 | 0.4 | 100.09 | 99.79 | 99.69 | 99.73 |
| mean | | | 99.83 | 100.11 | 99.98 | 99.88 |
| $\pm$ s.d. | | | $\pm$ 0.250 | $\pm$ 0.606 | $\pm$ 0.253 | $\pm$ 0.322 |
| $t$-test[a] | | | 0.367 | 0.251 | | |
| $F$-test[a] | | | 1.012 | 1.881 | | |
| HPLC method | | | 100.52 | 99.27 | | |
| | | | 100.81 | 99.51 | | |
| | | | 99.78 | 100.12 | | |
| mean | | | 100.37 | 99.55 | | |
| $\pm$ s.d. | | | $\pm$ 0.531 | $\pm$ 0.438 | | |
| $t$-test[a] | | | 0.762 | 0.655 | | |
| $F$-test[a] | | | 2.098 | 1.360 | | |

[a]Tabulated $t$- and $F$-values at $p = 0.05$ are 2.78 and 19.00, respectively.

## 6.4. Specificity

Also, the proposed methods were applied for the synchronized evaluation of LOS and AML in the co-formulated dosage form in ratio 10 : 1 for LOS and AML, respectively. There was no interfering from ordinary tablet excipients and additives. The results are shown in table 5. The results obtained were compared with those obtained by the comparison method [17] and there is no significant difference between them using $t$-test and variance ratio $F$-test.

# 7. Conclusion

Two accurate, sensitive, rapid and reproducible methods have been recognized for synchronized evaluation of LOS and AML in pure form and co-formulated tablets. The small values of RSD indicate the validity of the proposed methods; thus, the study allows their applications in quality control work.

Data accessibility. This article does not contain any additional data.
Authors' contributions. S.S. and J.J.N. carried out the laboratory work, data analysis, statistical analysis, the design of the study and drafted the manuscript. All authors gave final approval for publication.
Competing interests. We declare we have no competing interests.
Funding. No funding supported this research.

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
