## [Reviewer comments · Royal Society Open Science]

Review History

RSOS-181919.R0 (Original submission)

Review form: Reviewer 1

Is the manuscript scientifically sound in its present form?

Yes

Are the interpretations and conclusions justified by the results?

Yes

Is the language acceptable?

No

Is it clear how to access all supporting data?

Yes

Do you have any ethical concerns with this paper?

No

Have you any concerns about statistical analyses in this paper?

No

Recommendation?

Reject

Comments to the Author(s)

In the manuscript "Simultaneous Evaluation of Losartan and Amlodipine Besylate using Synchronous Spectrofluorimetric Technique and LC with Fluorimetric Detection ", the Authors described two independent analytical methods for the concertation determination of losartan (LOS) and amlodipine besylate (AML). The methods were optimized for the analysis of mixtures containing both compounds.

This manuscript is similar to previous publications of the same Authors (see for instance: Anal. Methods, 2015, 7, 8060, DOI: 10.1039/c5ay01134e) and "second derivative synchronous spectrofluorimetric" methods were already employed for the determination of drugs in their combined dosage form (see for instance: Luminescence. 2015 30(7):1011-9. doi: 10.1002/bio.2852). In addition, the HPLC method is not new.

Nevertheless, this is the first report on the application of the two methods to LOS and AML mixture analysis.

In general, the work is sound and the conclusions appear supported by the described results.

However, in my opinion, the language is not acceptable for a scientific publication. Consequently, the employed words are sometime misleading. Moreover, the text contains numerous typo errors and the format is non-homogenous (compare, for instance, the format of references 23, 24, and 25).

Additional points.

- 1) Several results are discussed in the text but not showed (see for instance paragraphs from 5.a.2 to 5.a.6). I suggest adding these results as supplemental materials.
- 2) Pag. 15, lines 20-23: "The results obtained by the suggested methods were matched with the results obtained from comparison method (17)." Please, add a brief description of the comparison method including the relevant experimental details.

Review form: Reviewer 2

Is the manuscript scientifically sound in its present form?

Yes

Are the interpretations and conclusions justified by the results?

Yes

Is the language acceptable?

Yes

Is it clear how to access all supporting data?

Not Applicable

Do you have any ethical concerns with this paper?

No

Have you any concerns about statistical analyses in this paper?

No

Recommendation?

Accept with minor revision (please list in comments)

Comments to the Author(s)

In page 2, line 32; The sentence is not clear. So, add "which" after "phase" and replace "carried out" by "pumped"

In page 3, line 10; the reference no. 1 should be taken from a pharmacopeia or a text book rather than a research paper.

Instead, the reference 1 research paper should be added to line 32 after "LOS"

The references 11 & 12 should be added in line 39 after "spectrophotometry" and TLC replaced by HPTLC.

In page 3, line 53; You should add "previously" after "were".

In page 5, line 6; the solvents used should be of HPLC grade.

Table 1 was not mentioned in the text.

Figures should be reordered as they are mentioned in the text so as figure 6 to be the last one.

In page 20, the format of reference 23 should be revised.

And, reference 24 title should be added.

Decision letter (RSOS-181919.R0)

18-Dec-2018

Dear Dr Shalan:

Manuscript ID: RSOS-181919

Title: "Simultaneous Evaluation of Losartan and Amlodipine Besylate using Synchronous Spectrofluorimetric Technique and LC with Fluorimetric Detection"

Thank you for submitting the above manuscript to Royal Society Open Science. Your paper was sent to reviewers and their comments are included at the bottom of this letter.

In view of the concerns raised by the reviewers, the manuscript has been rejected in its current form. However, a new manuscript may be submitted which takes into consideration these comments.

Please note that resubmitting your manuscript does not guarantee eventual acceptance, and that your resubmission will be subject to peer review before a decision is made.

Your resubmitted manuscript should be submitted by 17-Jun-2019. If you are unable to submit by this date please contact the Editorial Office.

On behalf of the Subject Editor Professor Anthony Stace and the Associate Editor Dr Ya-Wen Wang

REVIEWER(S) REPORTS:

Associate Editor Comments to Author ():

RSC Associate Editor:

Comments to the Author:

(There are no comments.)

RSC Subject Editor:

Comments to the Author:

(There are no comments.)

Reviewers' Comments to Author:

Reviewer: 1

Comments to the Author(s)

In the manuscript "Simultaneous Evaluation of Losartan and Amlodipine Besylate using Synchronous Spectrofluorimetric Technique and LC with Fluorimetric Detection", the Authors described two independent analytical methods for the concertation determination of losartan (LOS) and amlodipine besylate (AML). The methods were optimized for the analysis of mixtures containing both compounds.

This manuscript is similar to previous publications of the same Authors (see for instance: Anal. Methods, 2015, 7, 8060, DOI: 10.1039/c5ay01134e) and "second derivative synchronous spectrofluorimetric" methods were already employed for the determination of drugs in their combined dosage form (see for instance: Luminescence, 2015 30(7):1011-9. doi: 10.1002/bio.2852).

In addition, the HPLC method is not new.

Nevertheless, this is the first report on the application of the two methods to LOS and AML mixture analysis.

In general, the work is sound and the conclusions appear supported by the described results.

However, in my opinion, the language is not acceptable for a scientific publication. Consequently, the employed words are sometime misleading. Moreover, the text contains numerous typo errors and the format is non-homogenous (compare, for instance, the format of references 23, 24, and 25).

Additional points.

1) Several results are discussed in the text but not showed (see for instance paragraphs from 5.a.2 to 5.a.6). I suggest adding these results as supplemental materials.

2) Pag. 15, lines 20-23: "The results obtained by the suggested methods were matched with the results obtained from comparison method (17)." Please, add a brief description of the comparison method including the relevant experimental details.

Reviewer: 2

Comments to the Author(s)

In page 2, line 32; The sentence is not clear. So, add "which" after "phase" and replace "carried out" by "pumped"

In page 3, line 10; the reference no. 1 should be taken from a pharmacopeia or a text book rather than a research paper.

Instead, the reference 1 research paper should be added to line 32 after "LOS"

The references 11 & 12 should be added in line 39 after "spectrophotometry" and TLC replaced by HPTLC.

In page 3, line 53; You should add "previously" after "were".

In page 5, line 6; the solvents used should be of HPLC grade.

Table 1 was not mentioned in the text.

Figures should be reordered as they are mentioned in the text so as figure 6 to be the last one.

In page 20, the format of reference 23 should be revised.

And, reference 24 title should be added.

Author's Response to Decision Letter for (RSOS-181919.R0)

See Appendix A.

RSOS-190310.R0

Review form: Reviewer 1

Is the manuscript scientifically sound in its present form?

Yes

Are the interpretations and conclusions justified by the results?

Yes

Is the language acceptable?

No

Is it clear how to access all supporting data?

Yes

Do you have any ethical concerns with this paper?

No

Have you any concerns about statistical analyses in this paper?

No

Recommendation?

Accept with minor revision (please list in comments)

Comments to the Author(s)

The Authors addressed my concerns from the first round. However, in my opinion, an extensive editing for language is still needed.

Review form: Reviewer 2

Is the manuscript scientifically sound in its present form?

Yes

Are the interpretations and conclusions justified by the results?

Yes

Is the language acceptable?

Yes

Is it clear how to access all supporting data?

Yes

Do you have any ethical concerns with this paper?

No

Have you any concerns about statistical analyses in this paper?

No

Recommendation?

Accept as is

Comments to the Author(s)

- Dear Authors

The manuscript is well organised and I recommend it to be published

Decision letter (RSOS-190310.R0)

01-Mar-2019

Dear Dr shalan:

Title: Simultaneous Evaluation of Losartan and Amlodipine Besylate using Synchronous Spectrofluorimetric Technique and LC with Flourimetric Detection
Manuscript ID: RSOS-190310

Thank you for submitting the above manuscript to Royal Society Open Science. On behalf of the Editors and the Royal Society of Chemistry, I am pleased to inform you that your manuscript will be accepted for publication in Royal Society Open Science subject to minor revision in accordance with the referee suggestions. Please find the reviewers' comments at the end of this email.

The reviewers and handling editors have recommended publication, but also suggest some minor revisions to your manuscript. Therefore, I invite you to respond to the comments and revise your manuscript.

Please also include the following statements alongside the other end statements. As we cannot publish your manuscript without these end statements included, if you feel that a given heading is not relevant to your paper, please nevertheless include the heading and explicitly state that it is not relevant to your work. We have included a screenshot example of the end statements for reference.

- Ethics statement

Please clarify whether you received ethical approval from a local ethics committee to carry out your study. If so please include details of this, including the name of the committee that gave consent in a Research Ethics section after your main text. Please also clarify whether you received informed consent for the participants to participate in the study and state this in your Research Ethics section.

OR

Please clarify whether you obtained the necessary licences and approvals from your institutional animal ethics committee before conducting your research. Please provide details of these licences and approvals in an Animal Ethics section after your main text.

OR

Please clarify whether you obtained the appropriate permissions and licences to conduct the fieldwork detailed in your study. Please provide details of these in your methods section.

- Acknowledgements

Because the schedule for publication is very tight, it is a condition of publication that you submit the revised version of your manuscript before 10-Mar-2019. Please note that the revision deadline will expire at 00.00am on this date. If you do not think you will be able to meet this date please let me know immediately.

To revise your manuscript, log into <https://mc.manuscriptcentral.com/rsos> and enter your Author Centre, where you will find your manuscript title listed under "Manuscripts with Decisions". Under "Actions," click on "Create a Revision." You will be unable to make your

revisions on the originally submitted version of the manuscript. Instead, revise your manuscript and upload a new version through your Author Centre.

Best wishes,

Dr Laura Smith
Publishing Editor, Journals

RSC Associate Editor
Comments to the Author:
(There are no comments.)

Reviewer comments to Author:
Reviewer: 1

Comments to the Author(s)
The Authors addressed my concerns from the first round. However, in my opinion, an extensive editing for language is still needed.

Reviewer: 2

Comments to the Author(s)
- Dear Authors

The manuscript is well organised and I recommend it to be published

Author's Response to Decision Letter for (RSOS-190310.R0)

See Appendix B.

Decision letter (RSOS-190310.R1)

19-Mar-2019

Dear Dr Shalan:

Title: Simultaneous Evaluation of Losartan and Amlodipine Besylate using Synchronous Spectrofluorimetric Technique and LC with Fluorimetric Detection
Manuscript ID: RSOS-190310.R1

It is a pleasure to accept your manuscript in its current form for publication in Royal Society Open Science. The chemistry content of Royal Society Open Science is published in collaboration with the Royal Society of Chemistry.

RSC Associate Editor
Comments to the Author:
(There are no comments.)

Reviewer(s)' Comments to Author:

Appendix A

Response to the reviewers

In the manuscript “Simultaneous Evaluation of Losartan and Amlodipine Besylate using Synchronous Spectrofluorimetric Technique and LC with Fluorimetric Detection ”

- 1- This manuscript is similar to previous publications of the same Authors (see for instance: Anal. Methods, 2015, 7, 8060, DOI: 10.1039/c5ay01134e) and “second derivative synchronous spectrofluorimetric” methods were already employed for the determination of drugs in their combined dosage form (see for instance: Luminescence. 2015 30(7):1011-9. doi: 10.1002/bio.2852).

√-In this manuscript we use binary mixture[LOS and AML] and this is another combination other than [Valsartan and amlodipine] in Anal. Methods, 2015, 7, 8060, DOI: 10.1039/c5ay01134e or in[nebivolol and amlodipine] in Luminescence. 2015 30(7):1011-9. doi: 10.1002/bio.2852), with different experimental parameters

- 2- However, in my opinion, the language is not acceptable for a scientific publication. Consequently, the employed words are sometime misleading. Moreover, the text contains numerous typo errors and the format is non-homogenous (compare, for instance, the format of references 23, 24, and 25).

√-The corrections are done.

- 3- Several results are discussed in the text but not showed (see for instance paragraphs from 5.a.2 to 5.a.6). I suggest adding these results as supplemental materials.

√-The results were shown in supplementary materials

- 4- Pag. 15, lines 20-23: “The results obtained by the suggested methods were matched with the results obtained from comparison method (17).” Please, add a brief description of the comparison method including the relevant experimental details.

√-The correction are done

- 5- n page 2, line 32; The sentence is not clear. So, add “which” after “phase” and replace “carried out” by “pumped”

√-The correction are done

6- The references 11 & 12 should be added in line 39 after “spectrophotometry” and TLC replaced by HPTLC.

√-The correction are done

7- n page 3, line 53; You should add “previously” after “were”.
In page 5, line 6; the solvents used should be of HPLC grade.

√-The correction are done

8- Table 1 was not mentioned in the text.

√- Table 1 is mentioned under choice and optimization of chromatographic conditions

9- Figures should be reordered as they are mentioned in the text so as figure 6 to be the last one.

In page 20, the format of reference 23 should be revised.

And, reference 24 title should be added.

The correction is done

Appendix B

The Authors addressed my concerns from the first round. However, in my opinion, an extensive editing for language is still needed.

The language of the article is carefully revised and edited.